# Glycogen Synthase Kinase 3: Ion Channels, Plasticity, and Diseases

**DOI:** 10.3390/ijms23084413

**Published:** 2022-04-16

**Authors:** Mate Marosi, Parsa Arman, Giuseppe Aceto, Marcello D’Ascenzo, Fernanda Laezza

**Affiliations:** 1Department of Pharmacology & Toxicology, University of Texas Medical Branch, Galveston, TX 77555, USA; mgmarosi@utmb.edu (M.M.); paarman@utmb.edu (P.A.); 2Department of Neuroscience, Università Cattolica del Sacro Cuore, 00168 Roma, Italy; giuseppe.aceto1@unicatt.it; 3Fondazione Policlinico Universitario A. Gemelli, Istituto di Ricovero e Cura a Carattere Scientifico, 00168 Roma, Italy; 4Center for Addiction Research, Center for Biomedical Engineering, Mitchell Center for Neurodegenerative Diseases, Department of Pharmacology & Toxicology, 301 University Boulevard, Galveston, TX 77555, USA

**Keywords:** voltage-gated ion channels, sodium- and potassium-current, GSK3β-mediated phosphorylation, intracellular fibroblast growth factors (iFGF), protein–protein interaction, neuronal excitability, neuronal plasticity, neuropsychiatric disorders

## Abstract

Glycogen synthase kinase 3β (GSK3) is a multifaceted serine/threonine (S/T) kinase expressed in all eukaryotic cells. GSK3β is highly enriched in neurons in the central nervous system where it acts as a central hub for intracellular signaling downstream of receptors critical for neuronal function. Unlike other kinases, GSK3β is constitutively active, and its modulation mainly involves inhibition via upstream regulatory pathways rather than increased activation. Through an intricate converging signaling system, a fine-tuned balance of active and inactive GSK3β acts as a central point for the phosphorylation of numerous primed and unprimed substrates. Although the full range of molecular targets is still unknown, recent results show that voltage-gated ion channels are among the downstream targets of GSK3β. Here, we discuss the direct and indirect mechanisms by which GSK3β phosphorylates voltage-gated Na^+^ channels (Na_v_1.2 and Na_v_1.6) and voltage-gated K^+^ channels (K_v_4 and K_v_7) and their physiological effects on intrinsic excitability, neuronal plasticity, and behavior. We also present evidence for how unbalanced GSK3β activity can lead to maladaptive plasticity that ultimately renders neuronal circuitry more vulnerable, increasing the risk for developing neuropsychiatric disorders. In conclusion, GSK3β-dependent modulation of voltage-gated ion channels may serve as an important pharmacological target for neurotherapeutic development.

## 1. Introduction

Glycogen synthase kinase 3 (GSK3) is an evolutionarily conserved enzyme identified over 40 years ago in rabbit skeletal muscle as a negative regulator of glycogen synthesis [1,2]. As a serine/threonine protein kinase, GSK3 phosphorylates the first serine (Ser) and/or threonine (Thr) residue within the typical Ser/Thr_1_-X_1_-X_2_-X_3_-Ser/Thr_2_ motif. The first Ser/Thr residue within the motif is the GSK3 phosphorylation site, while the second Ser/Thr downstream residue serves as a priming site [3], typically phosphorylated by casein kinase II (CK2) [4,5,6]. Although substrate priming is not absolutely required for GSK3 phosphorylation, there is a 1000-fold increase in efficacy for primed targets [7,8]. Since its initial discovery, two isoforms—GSK3α and GSK3β—have been identified [9]. GSK3α and GSK3β share 85% similarity in amino acid sequence (98% similarity within the catalytic domain) but vary in pattern distribution and functional expression during development and adulthood [10]. Unlike other kinases, GSK3 is constitutively active in quiescent cells under basal conditions, and its activity is mainly controlled by inhibition. Extracellular stimuli such as growth factors, insulin, neurotransmitters, or reactive oxygen species (ROS) initiate intracellular signaling pathways that cause a reduction in GSK3β activity through a rapid and reversible N-terminal Ser9 phosphorylation (Ser21 for GSK3α) [10,11,12]. The phosphorylated N-terminus serves as a primed pseudo-substrate that folds into the positively charged pocket within the catalytic groove, occupying the active site of the enzyme via intramolecular interactions. This folding event prevents genuine substrates from entering the catalytic groove, thus blocking phosphorylation [8,13,14,15]. Multiple kinases such as protein kinase A (PKA), protein kinase B/Akt (PKB/Akt), and protein kinase C (PKC) can mediate Ser9 phosphorylation of GSK3β (Ser21 for GSK3α), leading to inhibition of the kinase [10,14,16,17,18,19,20]. Although less well-studied, positive regulation of GSK3β has also been demonstrated to occur in unstimulated, resting cells through Tyr216 phosphorylation (Tyr279 GSK3α) [20,21,22]. This phosphorylation increases enzymatic activity 5-fold [23], which may indicate that this phosphorylation event is more modulatory than activating. Reducing substrate affinity also occurs via dephosphorylation by protein tyrosine phosphatase—e.g., protein phosphatase 1 (PP1), protein phosphatase 2 A (PP2A), and protein phosphatase 2B (PP2B) [21]. Phosphorylation of Tyr216 in GSK3β (Tyr279 in GSK3α) induces a rotation and an upward movement of its amino acid side chain away from the substrate-binding groove, enabling genuine substrate accessibility [24,25,26]. It is important to note that inhibition by Ser9 phosphorylation can override activation by Tyr216 phosphorylation [27]. The exact mechanism of Tyr216/279 phosphorylation is not yet clear. However, experimental evidence suggests that it can happen either during the process of GSK3 translation, as a mechanism of intramolecular autophosphorylation, or as a post-translational modification induced by tyrosine kinases such as Fyn, Pyk2, and Csk [21,28,29,30,31].

Expression of GSK3β is highest in the central nervous system (CNS) [9,32,33], where it has roles in regulating gene transcription, metabolism, apoptosis, and cytoskeletal dynamics induced by substrate phosphorylation [10]. Not surprisingly, dysfunction of GSK3β has been implicated in a wide array of neuropsychiatric and neurological disorders, including mood disorder [34,35,36,37,38], schizophrenia [39,40,41,42,43], Alzheimer’s disease [44,45,46,47], Parkinson’s disease [48,49], and substance use disorder [50], raising a tremendous interest in targeting this kinase and its signaling pathways for neurotherapeutic development. To that point, the current limitation in targeting GSK3β is pharmacological catalytic disruption, which could lead to nonspecific effects. 

Over 500 substrates have been predicted for GSK3 [51], 50 of which are protein targets in the CNS [19,26]. Only recently, though, voltage-gated ion channels such as voltage-gated Na^+^ (Na_v_) [52,53] and voltage-gated K^+^ (K_v_) [54,55,56] channels have emerged as GSK3β substrates [34,57]. Herein, we provide an overview of the mechanisms by which GSK3β mediates direct and indirect effects on Na_v_ and K_v_ channels with implications for neuronal excitability, plasticity, and ultimately vulnerability to neuropsychiatric disorders [34,53,56,58,59]. We hope our review will inspire novel neurotherapeutic design strategies targeting pivotal nodes of critical GSK3-dependent neuronal pathways to overcome the existing barriers in direct pharmacological approaches against this kinase.

## 2. GSK3β and Na_v_ Channels

Na_v_ channels are a family of transmembrane proteins composed of α-subunits (Na_v_1.1–1.9) and auxiliary β subunits (β1–β4) [60,61,62,63]. The α-subunit is formed by four domains. Each domain is composed of six transmembrane segments (S1–S6) that form the channel pore (S1–S4) and the voltage-sensing unit (S5–S6). In response to membrane depolarization, the pore-forming α-subunit changes conformation, allowing a rapid influx of Na^+^ ions, which mediate the rising phase of an action potential. The β subunits are known to modulate kinetics and cellular trafficking of Na^+^ channels. Additionally, other proteins such as calmodulin and intracellular fibroblast growth factors (iFGFs) have been shown to regulate various physiological aspects of Na_v_ channels [64,65,66,67,68,69,70,71,72]. In neurons, action potentials generally propagate down the axon, initiating synaptic transmission by triggering the release of neurotransmitters into the synaptic cleft [73,74,75]. Within neurons, action potentials can also propagate backward throughout the somatodendritic compartment, where they contribute to synaptic signal integration via temporal and spatial summation [76,77,78,79,80,81,82,83]. With dozens of phosphorylation sites identified intracellularly [84,85], evidence for direct phosphorylation of Na_v_ channels by protein kinases such as PKC, PKA, and CK2 have been known for a long time [86,87,88,89]. Surprisingly, direct and indirect phosphorylation of Na_v_ channels and their regulatory iFGFs by GSK3β has emerged recently and is discussed in the following sections [52,53,66,87,90].

### 2.1. Direct Evidence of GSK3 Activity on Na_v_ Channels—Direct Phosphorylation

#### 2.1.1. GSK3β-Mediated Direct Phosphorylation of Na_v_1.2

Na_v_1.2 (encoded by SCN2A) is widely expressed in the CNS [91], appearing during embryonic development and reaching its maximal level of expression during adulthood [92]. During development, Na_v_1.2 appears in early phases of the myelin maturation process in myelinated fibers and at nodes of Ranvier [93]. In adulthood, it is typically located in unmyelinated axons preferentially enriched in the proximal region of the axon initial segment (AIS), closer to the soma, and is detected in apical dendrites [81,91,93,94,95,96,97,98]. Na_v_1.2 is also found in the cortex, thalamus, globus pallidus, and hippocampus [91]. In the cerebellum, Na_v_1.2 is found in both Purkinje and granule cells [99,100,101]. In addition to facilitating action potential backpropagation [81], Na_v_1.2 also plays a key role in regulating action potential frequency by modulation of repolarization in concert with K_v_ channels and setting the forward action potential threshold [96].

The first evidence for modulation and direct phosphorylation of Na_v_1.2 by GSK3β was published by James and colleagues [52]. In this study, pharmacological inhibition (with GSK-3 Inhibitor XIII) or genetic silencing of GSK3β was shown to increase the amplitude of Na_v_1.2-encoded currents, while opposite phenotypes were observed upon overexpression of the kinase. GSK3β inhibition was found not to affect the total level of mRNA expression or stability of the Na_v_1.2 protein [52], ruling out translation or protein degradation as plausible mechanisms underlying Na_v_1.2 regulation by GSK3. Surface labeling analysis of chimeric constructs expressing various intracellular domains of the Na_v_1.2 channel revealed that pharmacological inhibition of GSK3 increases the Na_v_1.2 C-terminal tail surface level but has no significant effects on other Na_v_1.2 intracellular domains. Furthermore, complementary evidence from the same study showed in vitro phosphorylation of the Thr1966 (T1966) residue within the Na_v_1.2 C-terminal tail [52] by GSK3β, indicating that GSK3 interacts with Na_v_1.2 through its C-terminal tail. Although the molecular mechanism underlying GSK3β’s control over Na_v_1.2-mediated current [52] is not yet clear, phosphorylation of Thr1966 might promote channel internalization (Figure 1A). Although speculative, GSK3 could act in concert with the E3 ubiquitin ligases Nedd4 and Nedd4-2 (neural precursor cell expressed developmentally downregulated protein 4 and 4-2) through the PPXY^1975^ motif downstream of Thr1966, through a mechanism that has been shown to regulate internalization and recycling of Na_v_1.6 [102,103]. The proposed functional relationship between Na_v_1.2 and GSK3β-dependent pathways (GSK3β-mediated phosphorylation) is likely to play an important role in modifying neuronal excitability. The impact of phosphorylation by GSK3β on Na_v_1.2 (predicted to decrease Na^+^ current) [52] could either limit action potential backpropagation from the proximal AIS (axon initial segment) into the somatodendritic compartment [81] or result in a paradoxical increase in neuronal excitability, largely through the failure of action potential repolarization by reduced K_v_ activation between action potentials [96,104,105]. Overall, GSK3β modulation of Na_v_1.2 could modify the ability of neurons to integrate dendritic synaptic signals with an impact on neuronal plasticity, which, if aberrant, could lead to cell vulnerability.

#### 2.1.2. GSK3β-Mediated Direct Phosphorylation of Na_v_1.6

Na_v_1.6 (encoded by SCN8A) is abundantly expressed in the CNS in both excitatory and inhibitory neurons, including hippocampal pyramidal neurons, granule cells in the dentate gyrus, cortical pyramidal neurons, motor neurons, medium spiny neurons of the nucleus accumbens (NAc), Purkinje, and granule cells of the cerebellum [106] (see also in Allen Brain Atlas; [107]). That being said, Na_v_1.6 represents the principal Na_v_ channel isoform in adulthood [94,108,109]. A unique feature of Na_v_1.6, recently confirmed by live imaging [109], is its subcellular localization at the distal AIS and nodes of Ranvier (40–70-fold difference as compared with the soma and dendrites) [81,93,94,97,110,111]. With its distinct subcellular localization at the distal AIS, Na_v_1.6 plays a crucial role in setting the action potential initiation threshold and forward propagation, and action potential frequency. When compared with Na_v_1.2, Na_v_1.6 has a lower threshold for activation [81], likely accounting for the initiation site of forward action potentials.

Despite the enrichment in axonal compartments, Na_v_1.6 channels can also form somatic nanoclusters and non-nanocluster classes [112,113]. Although the exact function of these nanoclusters is unclear, there is some speculation that these clusters participate in voltage-dependent current regulation, voltage-gated K+ (K_v_) channel coupling, and enhanced signaling fidelity via proximity [112]. With the aid of single-particle tracking and photoactivation localization microscopy, studies have shown that Na_v_1.6 nanoclusters are maintained in the soma via an ankyrin-independent mechanism, which is a biologically distinct mechanism from that observed in the Na_v_1.6 axonal pool [112]. In addition, the density of Na_v_1.6 channels has been found to gradually decrease along the axo-somatic axis [113] and the proximal axis of the dendrites, without clear evidence of labeling in dendritic spines [111]. Although the physiological complexity of Na_v_1.6 is far from being resolved, the co-existence of Na_v_1.6 somatic, axonal, and dendritic pools indicates that this channel operates via mechanisms that are far more complex than previously hypothesized. Because of their localization at the AIS, it is even possible that Na_v_1.6 works in concert with Na_v_1.2 [81,94,98,114].

Na_v_ channels are heavily modulated by post-translational modifications. Following the identification of the Na_v_1.2^T1966^ residue as a substrate of GSK3β, evidence has been provided for a corresponding phosphorylation mechanism on the Na_v_1.6 C-terminal tail (Figure 1B). In Scala et al. (2018), GSK3β was reported to directly phosphorylate Na_v_1.6 at residue Thr1938 (Na_v_1.6^T1938^) [53]. In the same study, recombinant unphosphorylated GSK3β and the Na_v_1.6 C-terminal tail were found to form a stable complex via direct interaction using surface plasmon resonance. Additionally, when GSK3β was pharmacologically inhibited (by CHIR99021) or genetically silenced, Na_v_1.6-mediated currents were found decreased, and V^1/2^ of steady-state inactivation shifted toward a more hyperpolarized potential (~8 mV). Conversely, opposite phenotypes were found upon GSK3β overexpression. These findings are important for the following reasons: i. they establish a clear separation between the physiological outcome of GSK3β-dependent phosphorylation of Na_v_1.2^T1966^ and Na_v_1.6^T1938^—namely, that the corresponding phosphorylation on the two channels results in opposite phenotypes relative to Na^+^ currents; ii. they demonstrate the direct binding of GSK3β to the Na_v_1.6 C-terminal tail, independent of the phosphorylation activity, hinting at a previously undescribed role of GSK3β as a protein scaffold for the Na_v_1.6 channel complex; iii. lastly, the combination of effects induced by pharmacological inhibition or genetic silencing of GSK3β on the biophysical properties of the Na_v_1.6 channel (V^1/2^ of steady-state inactivation) indicates that the effect of GSK3β on Na_v_1.6 includes complex effects on the channel kinetics. In the same study, a decoy peptide competing with the GSK3β phosphorylation motif on the Na_v_1.6 C-terminal tail demonstrated that GSK3β-dependent phosphorylation of Na_v_1.6^T1938^ plays a role in the maladaptive firing of medium spiny neurons (MSN) in the NAc. This finding further broadened the implication for phosphorylation of Na_v_1.6 in the context of neural plasticity and could aid in the development of novel therapies for brain disorders (Alzheimer’s, Parkinson’s, Huntington’s, etc.) associated with either upregulation of GSK3β or Na_v_1.6 activity [115,116,117,118].

### 2.2. Indirect Effects of GSK3β Activity on Na_v_ Complex—Protein–Protein Interactions (PPI) 

Na_v_ channels form macromolecular complexes with a variety of intracellular regulatory proteins via stable or transient protein–protein interactions (PPIs) [72,119,120,121]. These PPIs determine the subcellular trafficking, compartmentalization, and functional expression of Na_v_ channels, while fine-tuning neuronal excitability [122,123,124,125,126,127].

Within this category are the intracellular fibroblast growth factors (iFGFs; FGF11–FGF14) that have been shown to form molecular complexes with the C-terminal tail of various Na_v_ isoforms, resulting in functional specialization of Na^+^ currents [68,71,128,129]. FGF14 is one of these complex-forming proteins. Inherited loss-of-function mutations in the FGF14 coding region disrupt the trafficking of Na_v_ α subunits to the AIS, attenuate Na_v_ current densities, and impair excitability of hippocampal neurons [128,130]—phenotypes that are thought to contribute to spinocerebellar ataxia 27 (SCA27) [129,131].

When studied in isolation in heterologous cells, FGF14 has been shown to produce changes in Na^+^ peak current amplitude, voltage-dependence of activation, and/or steady-state inactivation of Na_v_1.1, Na_v_1.2, and Na_v_1.6 channels in an isoform-specific manner [67,68,90,132,133,134,135,136]. Although high-resolution structures of FGF14 and any of the Na_v_ channel isoforms are not yet available, homology models derived from a closely related factor (FGF13) and its binding partners (Na_v_1.5 and calmodulin) [137,138] provide evidence for direct monomeric binding of FGF14 to the Na_v_ channel C-terminal tail [133,134,135,139,140]. In vivo equilibrium between the monomer and dimer forms has been postulated and could be critical for finely tuned modulation of neuronal excitability via JAK2 signaling [136]. In addition, there is also evidence of monomeric interactions between FGF14 and Na_v_1.6 from previous studies [132,135].

The Shavkunov, Wildburger, and Nenov paper (2013) utilized a combination of high-throughput screening (HTS) and orthogonal assays to screen a chemical library of kinase inhibitors that led to the first demonstration of the FGF14:Na_v_1.6 channel complex as a downstream target of GSK3β [90]. In this study, the PPI between FGF14 and Na_v_1.6 was reconstituted in cells using the split-luciferase complementation assay (LCA) [69,90,141]. Upon treatment with chemically diverse inhibitors of GSK3β or siRNA targeting GSK3β, FGF14–Na_v_1.6 channel complex formation was reduced (Figure 1B). Functionally, this resulted in modulation of Na_v_1.6 currents and led to dissociation of the FGF14–Na_v_ channel complex at the AIS, reduction in action potential threshold, and overall excitability [90,142]. Follow-up studies revealed that GSK3β phosphorylates FGF14 at Ser226 and that a silent Ala mutation at Ser226 reduces FGF14–Na_v_1.6 complex assembly [143]. Additionally, FGF14–Na_v_1.6 channel assembly, cellular distribution, and functional activity are regulated by CK2, the most common priming kinase for GSK3β. Pharmacological inhibition of GSK3 was also effective in decreasing the interaction between FGF14 and Na_v_1.2 and in reversing FGF14-dependent modulation of Na_v_1.2 currents, a mechanism that could be mediated in part by GSK3-dependent phosphorylation of FGF14 at Ser226 (Figure 1A) [90]. Inhibition of CK2 causes an array of molecular, cellular, and functional phenotypes that closely mirror those observed upon GSK3β inhibition and have been mechanistically linked to CK2 phosphorylation of S228 and S230, two residues that lie within the GSK3β/CK2 priming motif [87]. Overall, GSK3β exerts effects on Na_v_ channels that are not only mediated by direct phosphorylation of the channel C-terminal tail but occur through a cascade of regulatory events on FGF14 that, in turn, affect Na_v_ channels. 

### 2.3. Indirect Evidence of GSK3β Activity on Na_v_ Complex—Signaling Pathways

Extensive evidence indicates that GSK3β is also an integral part of the Wnt/β-catenin pathway and that GSK3β-dependent phosphorylation of β-catenin leads to proteasomal degradation of the complex [144]. GSK3β phosphorylates β-catenin at S33/S37/T41 [145], which connects β-catenin to a βTrCP E3 ligase degradation pathway. Stimulation of Wnt leads to an inhibition of GSK3β, which prevents the β-TrCP-mediated breakdown of β-catenin. Non-phosphorylated β-catenin accumulates in the cytoplasm, translocates to the nucleus, and initiates transcription of target genes [146,147,148,149]. It is also essential to note that β-catenin plays an important role in neuronal activity. It regulates neurogenesis, neuronal polarity together with ankyrin-G (localization of AIS) [150], dendritic morphogenesis [151], cell adhesion via N-cadherin [152,153], and transcription of several genes [145,154]. β-catenin is also known as a critical mediator of axonal growth in neurons [155,156], which is crucial for neuronal development, synaptic health, network organization, and proper brain development. Several studies have revealed a role for the GSK3β signaling pathway in regulating neuronal/axonal polarity [156,157,158] and the distribution of Na_v_ channels at the AIS, which overlap with the functions of β-catenin mentioned above. The study by Tapia and colleagues (2013) showed that knockdown of β-catenin or pharmacological inhibition of GSK3β (by GSK3 inhibitor X and AR-A014418) decreased the levels of phosphorylated-β-catenin, ankyrin-G, and Na_v_ channels at the AIS, leading to impaired neuronal excitability [156]. Additionally, inhibition of GSK3β has been shown to affect the number of AISs in cultured neurons, with effects on action potential initiation [159]. These studies are in line with the Shavkunov, Wildburger, Nenov et al., (2013) paper, demonstrating that pharmacological inhibition of GSK3 (GSK3 inhibitor XIII and CHIR99021) induces dissociation of FGF14 from the Na_v_ channels at the AIS and redistribution of the complex in the somatodendritic compartment [90]. As mentioned earlier, pseudo-substrate inhibition of GSK3β via Ser9 phosphorylation can be prevented with compounds such as triciribine, allowing increased GSK3β activity and ultimately increasing Na_v_1.6-mediated currents [160]. These results are consistent with the idea that GSK3β activity is directly connected to increased neuronal excitability. Another potential cross-talk between Na_v_ channel activity and GSK3β could arise from variation in the level of Wee1 kinase. GSK3β can modulate and degrade Wee1 through ubiquitination signaling [161,162,163]. Wee1 may also activate AKT (PkB), which, as mentioned earlier, can inactivate GSK3β through S9 pseudo-substrate phosphorylation [163]. Not only is the Wee1 kinase a crucial component of the G2-M cell cycle checkpoint, and its activity is modulated by GSK3 [164], but also it has been shown to confer isoform-specific effects on Na_v_ channels via PPIs regulation [165].

Further evidence for the involvement of the Wnt/β-catenin pathway in GSK3β activity is the cross-talk between GSK3β and Na_v_1.5 that has been shown in iPSC-derived cardiomyocytes. In patients with arrhythmogenic cardiomyopathy due to plakophilin 2 (PKP2) mutations, the Wnt/β-catenin pathway was decreased, and Na^+^ currents were found to be reduced. GSK3β inhibition and Wnt/β-catenin signaling activation restored the cardiomyocyte function and Na_v_ currents [166], suggesting that the Wnt/β-catenin/GSK3 pathway plays a role in the regulation of cardiomyocyte Na_v_ channels. Application of a selective GSK3β inhibitor and/or Wnt3a (recombinant protein) led to a reduction in the mRNA level of Scn5a, accompanied by a reduction in Na^+^ current density [167,168]. It is also known that Wnt/β-catenin in the presence of the Wnt ligand sequesters GSK3β from the cytosol [169,170], further supporting the connection between Wnt/β-catenin signaling activation and Ser9-independent GSK3β inhibition.

For neurons to make connections, axonal and dendritic structures must be organized appropriately. During this reorganization, cytoskeletal components such as microtubules and/or actin are utilized to localize and regulate Na_v_ channel function [171,172]. Microtubules are dynamic and require stability via phosphorylation of microtubule-associated proteins (MAPs) [173,174,175,176,177]. Many MAPs are substrates for GSK3β [174,176,177,178,179]. Therefore, a link is likely between GSK3β, Na^+^ channels, and cytoskeleton dynamics [26,180,181,182,183]. Additionally, inhibition of GSK3 prevents accumulation of the actin-related protein 2/3 complex (Arp2/3) in cellular extensions. Arp2/3 binds to the existing F-actin filament and initiates new filament creation in dendritic spines, which results in spine volume expansion [183,184,185,186].

GSK3β has also been shown to be recruited to the β1-integrin complex. Integrin-linked kinase (ILK) is known to connect the β1-integrin complex to the actin cytoskeleton. GSK3β is also a substrate for ILK, which inhibits GSK3 activity in a PI3K-dependent manner [187,188]. Other cytoskeletal components, such as β-adducins, are also known as GSK3β substrates [189].

Altogether, this evidence supports a link between GSK3β, Na_v_ channels, and neuronal cytoskeletal components during neuronal morphogenesis, cell morphological reorganization, cell motility, migration, and the transport of cargo (e.g., mitochondria) [181,183,190,191].

There is another potential cross-talk between GSK3β and Na_v_ channels (Na_v_1.7) [192]. In bovine adrenal chromaffin cells [193,194], veratridine-induced potentiation of Na_v_1.7-mediated currents [194,195,196] leads to a cascade of intracellular events that converge onto inactivation of GSK3β [195,197], mimicking the effect of GSK3 inhibitors [198]. Finally, veratridine-induced Na_v_1.7–Ca^2+^ influx decreases GSK3β-catalyzed Ser396-phosphorylation of tau; and activates ERK1/ERK2 and p38-primed Na_v_1.7 to increase Na^+^ influx. It is also known that chronic insulin-like growth factor-1 treatment upregulates cell surface expression of Na_v_1.7 by inhibition of GSK3β [199], causing an increased Na^+^ influx via the PI3K/Akt/GSK3β pathway. The increased Na^+^ influx (via upregulated cell surface Na_v_ channels) enhances Na^+^ influx-induced Ser9 phosphorylation/inhibition of GSK3β, which, in turn, upregulates cell surface Na_v_1.7 channels. This dual relationship between GSK3β and Na_v_1.7 may act as a positive feedback loop triggered by GSK3β inhibition and maintained by Na_v_1.7-mediated Na^+^ influx. 

## 3. GSK3β and K_v_ Channels

Among the different types of K^+^ channels, encoded by more than 80 genes, the K_v_ channels represent the largest and most complex family, classified into K_v_1–K_v_12 subfamilies [200]. Mammalian K_v_ channels are tetramers, composed of α-subunits that line the ion pore. Each α-subunit shows six α-helical transmembrane domains (S1–S6), a membrane-reentering P loop between S5 and S6, and cytosolic N-, C-termini. The S5-P-S6 segments constitute the ion conduction pore, while the S1–S4 sequences are critical for voltage-sensing and gating [201]. K_v_ channels regulate the rapid and selective exchange of K^+^ ions and produce electrical currents, which allow neurons to efficiently repolarize back to the resting membrane potential following an action potential. In addition to their fundamental role in neuronal excitability, K_v_ channels are known to substantially contribute to action potential backpropagation, with implications for synaptic transmission and plasticity [202,203]. Protein phosphorylation is the most common post-translational modification in signal transduction [204]. Indeed, there is abundant and increasing evidence that K_v_ channels are direct targets of several protein kinases and phosphatases, resulting in dynamic and reversible changes in K_v_ channel expression, localization, and functions [205]. Within this context, of all the different K_v_ channel subtypes, only K_v_7.2 and K_v_4.2 have been identified as substrates of GSK3β.

### 3.1. GSK3β and K_v_7.2 Channels

The K_v_7 family of voltage-gated K^+^ channels consists of five members (from K_v_7.1 to K_v_7.5) encoded by the KCNQ genes (KCNQ1–5). Within neurons, K_v_7.2 and K_v_7.3 channels exist as heteromeric and homomeric subunit assemblies that mediate M-currents—a hyperpolarizing current that reduces neuronal excitability [206]. M-current is a non-inactivating, slowly activating/deactivating current that is suppressed upon stimulation of M1 muscarinic receptors, as well as other Gq-coupled receptors. M-current activation hyperpolarizes the cell membrane, thus reducing neuronal excitability. M-current suppression depolarizes the neuron and increases neuronal excitability [206]. Moreover, M-current also has a significant role in subthreshold membrane oscillations in many types of neurons in the CNS [207,208,209].

The K_v_7.2 channel is phosphorylated by GSK3β in vitro [210]. In neurons, K_v_7.2 and GSK3β colocalize in apical dendrites of the pyramidal neurons in the medial prefrontal cortex (mPFC). At this level, administration of GSK3β inhibitors leads to increased excitability through M-type currents reduction [58]. Consequently, GSK3β inhibitors in the mPFC cause reduced prepulse inhibition [58], which reflects a dysfunction in sensorimotor gating—a feature reported in patients with psychiatric disorders such as bipolar disorder and schizophrenia [211]. Accordingly, mice with overactive GSK3β show dysfunction in sensorimotor gating, as measured by increased startle responses. Altogether, these findings indicate that GSK3β inhibition of the K_v_7.2 channel may cause hyperexcitability in psychiatric diseases (Figure 1C). 

As described above, the phosphorylation of certain residues of K_v_7.2 channels is known to cause suppression in channel activity, but little is known about the mechanism of dephosphorylation. One phosphatase that plays a crucial role is the protein phosphatase 2A (PP2A). The experimental results of Borsotto et al. (2007) suggest that PP2A, via its Bγ-regulatory subunit, stimulates the dephosphorylation of K_v_7 and increases channel activity by counteracting the inhibiting effects of GSK3β-mediated phosphorylation [34,210,212]. Dysfunction of the KCNQ2/GSK3β/PP2A-Bγ regulation pathway, and its balance, can explain changes in neuronal excitability that underline the manic and depressive episodes occurring in mood disorders (e.g., bipolar affective disorder) [210].

Furthermore, GSK3β may also affect K_v_7.2 through FGF14-dependent phosphorylation. In fact, recent research has shown that FGF14 interacts with KCNQ2 in the AIS, thus bridging Na_v_1.6 and KCNQ2. This finding implicates FGF14 as an organizer of channel localization in the AIS and suggests possible GSK3β/FGF14-dependent modulation of K_v_7.2 conductance [213]. Since M-channel dysfunction has been linked to schizophrenia [214], epilepsy [215,216,217], and bipolar disorder [210,218,219], neuronal hyperexcitability resulting from impaired GSK3β-dependent M-channel dysfunction is likely a common denominator in several neurological and psychiatric illnesses [220,221,222,223]. 

### 3.2. GSK3β and K_v_4.2 Channels

Subthreshold-activating, rapid inactivating A-type K^+^ currents are essential for the proper functioning of neurons, where they act as key regulators in AP repolarization, repetitive firing, and backpropagation of AP [224,225,226]. In addition, the gating property and distribution of A-type K^+^ channels influence intracellular Ca^2+^ levels in dendritic branches [227]. Because Ca^2+^ is fundamental for many forms of synaptic plasticity in dendrites, the regulatory effects of A-type K^+^ channels on dendritic excitability are critical for dendritic and synaptic processing during synaptic plasticity [227,228].

Although several K^+^ channel pore-forming subunits can generate A-type currents, including K_v_1 and K_v_3 subfamilies, results from many studies indicate that members of the K_v_4 subfamily are the main determinants of A-type K^+^ currents in somatodendritic regions of neurons [229]. Among the three K_v_4 genes (K_v_4.1–3), K_v_4.2 is prominently expressed in many brain regions, including the hippocampus, cerebral cortex, and NAc [55,230]. K_v_4.2 channels are clustered in macromolecular complexes with auxiliary subunits, including K^+^ channel-interacting proteins (KChIP1–4) and dipeptidyl peptidases 6 and 10 (DPP6 and DPP10) [231]. Both KChIPs and DPPs work together to modify K_v_4.2 expression, membrane surface localization, and channel kinetics. Most of the modifications to K_v_4.2 channel localization and function have been attributable to several protein kinase phosphorylations [80,232,233]. The phosphorylation of K_v_4.2 by protein kinase A (PKA), protein kinase C (PKC), and extracellular signal-regulated kinase/mitogen-activated protein kinase (ERK/MAPK) result in downregulation of the currents [234]. In hippocampal pyramidal neurons, this downregulation facilitates an increase in somatodendritic excitability, enhancing susceptibility to network hyperexcitability. The association of the K_v_4.2 A-type K^+^ channel and its auxiliary subunits—KChIP, DPP6, and DPP10—are also regulated by protein kinases [232,233,235,236].

In addition to PKA-, PKC-, and ERK/MAPK-mediated phosphorylation of K_v_4.2, recent findings support the notion that GSK3β is also a key modulator of K_v_4.2 channels [55,56,237]. Indeed, Ser-616 of K_v_4.2 lies in the GSK3β phosphorylation motif since it is flanked by Pro617 and followed by Ser620.

The first observation that Ser 616 of K_v_4.2 is a phosphorylation site of GSK3β was obtained in cultured hippocampal neurons [237]. Treatment of neurons with amyloid-β protein (Aβ42) induces a decrease in A-type K^+^ currents and an increase in excitability through a GSK3β-dependent mechanism, an effect that was paralleled with phosphorylation of K_v_4.2 at Ser-616 [237]. In a subsequent study, the ability of GSK3β to phosphorylate K_v_4.2 subunits through a direct mechanism was confirmed in human embryonic kidney (HEK) 293 cells expressing exogenous K_v_4.2 subunits [56].

Further support for GSK3β–K_v_4.2 interaction was demonstrated in both cortical somatosensory pyramidal neurons and MSNs of the NAc [55,56]. In particular, pharmacological and genetic downregulation of GSK3β levels resulted in an increased A-type K^+^ current density. Conversely, positive pharmacological modulation of GSK3β levels (by CHIR-99021, also known as CT99021) had the opposite effect on A-Type K^+^ current density [55,56] (Figure 1D).

## 4. GSK3β: A Central Node for Plasticity

The role of GSK3β-dependent pathways in the induction and expression of activity-dependent plasticity is supported by extensive experimental evidence. Pioneering studies suggest that inhibition of GSK3 is critical in facilitating the induction of long-term potentiation (LTP), while increased activity promotes long-term depression (LTD) and controls metaplasticity in the hippocampal region [45,238,239]. These observations were followed by more recent investigations supporting mechanistic links between GSK3β inhibition (by Ser9 phosphorylation via Akt/PKB) and LTP in the dentate gyrus [240,241]. Molecularly, GSK3β-mediated plasticity has been associated with NMDA and AMPA receptor trafficking, dopamine receptor 2 (DR2)-DISC1 interactions, and epigenetic regulation of NMDA receptor expression and function, as well as changes in synaptic spine density and post-synaptic GABA receptor microdomains [242,243,244]. Interestingly, impairments in synaptic plasticity due to GSK3 activity disruption have been demonstrated to occur upon pharmacological manipulation. On the other hand, GSK3 knock-in mice carrying S9 and/or S21 Ala mutations, with a constitutively hyperactive kinase, exhibit LTP/LTD modifications. Due to the critical roles of GSK3 in synaptic plasticity, it is not surprising that the dynamic regulation of this kinase has been linked to many circuitry functions. In addition, studies have implicated GSK3 in synaptic weakening during the cortical oscillations of slow-wave sleep, cognitive performance, and fear conditioning, with mechanistic links in Alzheimer’s disease, bipolar disorder, schizophrenia, and Fragile X syndrome [38,40,241,245,246].

Surprisingly, the molecular mechanisms by which these forms of GSK3-dependent plasticity are established at the neuronal, circuitry, and behavioral levels are still poorly understood. The recent identification of Na_v_ and K_v_ channels as direct targets of GSK3 phosphorylation has opened new horizons in investigating the potential mechanisms underlying neuronal plasticity, with implications for advancing therapeutic development against neuropsychiatric disorders. In the next section, we discuss the evidence for GSK3β-mediated fine-tuning of Na_v_ and K_v_ channels and its putative impact on neuronal plasticity and behavior.

### 4.1. Potential Mechanisms by Which GSK3β Influences Neuronal Excitability/Plasticity

Changes to pre- and postsynaptic neuronal dynamics are critical components of plasticity. By influencing the activity of Na_v_ and K_v_ channels, GSK3β could serve as a central node that determines and/or modifies neuronal activity by modulating both pre- and post-synaptic output. The modifying effects of GSK3β on Na_v_ and K_v_ channel-mediated currents are expected to be crucial for the generation and propagation of action potentials [247,248,249]. In neurons, the fire rate, pattern, directionality, and threshold of action potentials are determined by the location, distribution, kinetics, and expression level of Na_v_ and K_v_ channels [98,113,250]. Ultimately, the ensemble of functionally active Na_v_ and K_v_ channels in axons, dendrites, and somas will determine the likelihood of forward vs. backpropagation of action potentials, which, in turn, affects synaptic transmission at presynaptic axonal terminals (forward propagation) and synaptic input integration (backward propagation) in the dendritic tree [251,252,253].

Changes in the activity of GSK3β can modulate functional properties of Na_v_1.2, Na_v_1.6, K_v_7.2, and K_v_4.2 channels directly via phosphorylation or indirectly via PPIs (Figure 1). In some cases, this can lead to the opposite effects. Intriguingly, while increased GSK3β activity suppresses Na_v_1.2 [52], K_v_4.2, and K_v_7.2 currents [56,58,254], it increases Na_v_1.6 currents [53]. These isoform-specific effects on Na_v_ channels, complemented with the reported changes induced by GSK3β on K_v_ currents, could greatly favor forward action potential propagation in mature (myelinated) neurons expressing predominantly Na_v_1.6. On the other hand, during early developmental stages when neurons abundantly express Na_v_1.2, increased activity of GSK3β could favor backpropagating action potentials by suppressing Na_v_1.2 and K_v_ currents, thus increasing the computational ability of neurons to integrate synaptic inputs in the dendritic tree. Parallel with the physiological effects, it is important to note that inhibition of GSK3β significantly decreased the length of the β-IV spectrin-positive AIS in cultured hippocampal neurons [142], which further supports the regulation of axonal function by this kinase. 

Ultimately, the posttranslational modifications of Na_v_ and K_v_ channels are expected to affect network-level functions involved in a variety of different forms of plasticity—e.g., STDP, LTD/LTP, and memory formation [57,239,255] and related behaviors. Understanding how GSK3β-mediated phosphorylation affects the above-mentioned networks and behaviors can give insight into the maladaptive plasticity events that occur in various disease states (e.g., addiction, mood disorders, neurodegeneration, epilepsy, and sleep dysregulations).

### 4.2. GSK3β Influences Neuronal Excitability—Maladaptive Behavioral Outcomes

#### 4.2.1. Isolated vs. Enriched Environmental Outcomes

Scala and colleagues identified GSK3β and Na_v_1.6 as regulators of neuronal plasticity induced by environmentally enriched (EC) or isolated conditions (IC) [53]. EC/IC protocol is a non-drug, non-genetic, and non-surgical model of inducing resilience or vulnerability to reward in the context of neuropsychiatric disorders. An enriched environment produces resilience to addiction-related and depression-like behaviors in rodents, while isolated conditions promote vulnerability [256,257,258,259].

MSNs of the NAc are known to be associated with depression-like behaviors, addiction, and neurodegeneration [260,261,262,263,264]. Transcriptomic analysis (ingenuity pathway-bioinformatic analysis) of NAc showed a significant decrease in the canonical PI3K/Akt/GSK3β signaling pathway. Furthermore, decreased levels of mRNA coding for GSK3β and Na_v_1.6 channels (SCN8A) were found in resilience rats (EC condition), compared with vulnerable rats (IC condition) [53]. Under isolated conditions, GSK3β is expected to be hyperactive (e.g., increased Tyr216 phosphorylation or mRNA production), leading to an increase in firing frequency in MSN. Scala and colleagues (2018) proved that MSNs have an elevated firing rate in IC rats [53]. The changes in GSK3β level/activity and its effect on Na_v_1.6-mediated current (increased sodium current) could be why the IC (vulnerable rats) rats exhibited a prominent increase in neuronal excitability (increased amplitude of *I*_NaP_ and AP firing rate) compared with MSNs from EC (resilience) rats [53]. This may indicate that enriched environmental conditions reduce MSNs output by reducing GSK3β activity at the circuit level, thus inhibiting maladaptive behavior.

Similar results were found in GSK3-KI animals: in the SK3α21A/21A/β/9A/9A knock-in mouse, the lack of GSK3 inhibitory phosphorylation by PKB/Akt kinase results in constitutively high GSK3 activity. This animal model mimics behavioral traits of mood disorders (e.g., depression) [265,266]. In MSNs of GSK3-KI mice, both *I*_NaP_ and firing increased significantly compared with wild-type controls [53]. Maladaptive neuronal firing developed in IC rats and GSK3-KI animals can be prevented by pharmacological inhibition of GSK3β (by CHIR99021). Additionally, in vivo genetic silencing (AAV-shGSK3β) or a decoy peptide encompassing the GSK3β phosphorylation site on Nav1.6 prevented maladaptive firing in IC rats. These findings confirm that the GSK3β–Na_v_1.6 complex contributes to maintaining maladaptive firing in MSNs [53].

Based on these findings, changes in the intrinsic excitability could be an early marker of vulnerability in response to a variety of conditions that have been associated with the vulnerability of neurons in the NAc, such as alcohol and drug abuse [267,268], chronic stress, and prolonged social isolation [257,264]. Overall, direct GSK3β-mediated phosphorylation of voltage-gated ion channels and/or indirect phosphorylation of accessory proteins affects neuronal firing rate and plasticity, with potentially critical implications for behavior. In the next section, we describe the findings from another animal model in which the environment is modified to determine its effects on behavior.

#### 4.2.2. Chronic Unpredictable Mild Stress Evoked Maladaptive Plasticity Outcomes

GSK3β-dependent modulation of the K_v_4.2 channel has been recently identified as a novel regulatory mechanism of spike-timing-dependent plasticity (STDP) in layer 2/3 pyramidal neurons of the somatosensory cortex [56], a form of activity-dependent remodeling that is induced by a moderate number of repeated timed activations of pre- and postsynaptic neurons [269]. Beyond the physiological role of the GSK3β/K_v_4.2 axis in the regulation of STDP, recent evidence indicates that this molecular pathway plays a critical role also in determining maladaptive plasticity of NAc MSNs in a chronic unpredictable mild stress (CUMS) mouse model [55].

The CUMS model is a nondrug, nonsurgical-mediated paradigm that confers depression-like behaviors in rodents. Although changes in signaling pathways, neuroplasticity, and neuroinflammation are involved in developing depression-like behaviors, strong evidence indicates that maladaptive plasticity in NAc circuitry plays a fundamental role in establishing these behaviors [270]. In accordance with the general notion of increased GSK3β signaling in the CUMS model [271], Aceto and colleagues found a significant reduction in inhibitory phosphorylation at Ser-9 of GSK3β in the NAc of CUMS mice [55]. In MSNs from these mice, spike-timing-dependent LTP (tLTP) is increased and normalized by pharmacological inhibition (by CHIR99021) or gene silencing of GSK3β. Furthermore, knockdown of GSK3β in the NAc ameliorated depressive-like behavior in CUMS mice. Consistent with the involvement of the GSK3β/K_v_4.2 axis in maladaptive plasticity in the CUMS model of depression-like behavior, the increased tLTP found in these mice has been associated with complementary modulation (decrease) in K_v_4.2-mediated A-type K^+^ currents [55]. 

Overall, these studies highlight the critical role of the GSK3β–K_v_4.2 complex in the context of MSN maladaptive plasticity occurring in mouse models of depression. Moreover, the finding that GSK3β knockdown prevents CUMS-induced depressive-like behavior paves the way to consider the GSK3β–K_v_4.2 axis as a new therapeutic target for managing the negative impact of chronic stress.

## 5. Conclusions and Future Directions

As we discussed above, disruption of GSK3β activity could be a risk factor for neuropsychiatric and neurodegenerative disorders [43,53,55,143,272]. A possible target for pharmacological therapy is harnessing the power of direct and indirect modulation of Na_v_ and K_v_ channels by GSK3β. We have collected the supporting evidence of GSK3β-mediated phosphorylation of Na_v_1.2, Na_v_1.6, K_v_4.2, and K_v_7.2 channels (Figure 1 and Figure 2). These ion channels are essential for fine-tuning neuronal excitability and synaptic transmission. Any disruption of their functionality could be a starting point of neuronal dysfunction, maladaptive plasticity, and behaviors that could lead to complex brain disorders. Thus, pharmacological modulation of specific targets of the GSK3β pathway could potentially be valuable as therapy management for a variety of different neuropsychiatric and neurodegenerative disorders.

## Figures and Tables

**Figure 1 ijms-23-04413-f001:**
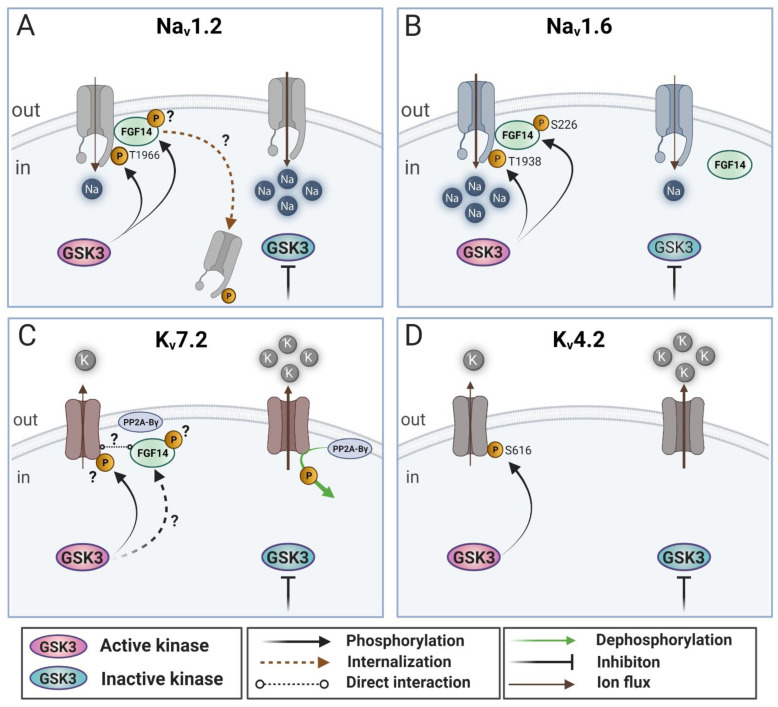
GSK3β-mediated phosphorylation as a regulation of voltage-gated cation channels. (**A**) Effects of GSK3β-mediated phosphorylation on the Na_v_1.2 channel. Active GSK3β phosphorylates the Na_v_1.2 channel at the Th1966 (T1966) residue, which maybe facilitates channel internalization (dotted line) and causes a decrease in the Na_v_1.2-mediated Na^+^ current and in neuronal excitability. Additionally, active GSK3β stabilizes the FGF14:Na_v_1.2 channel complex assembly, a phenotype that could be in part mediated by phosphorylation of FGF14. (**B**) Active GSK3β phosphorylates the Na_v_1.6 channel at the Th1966 (T1938) residue (direct modulation of Na_v_1.6), and FGF14 at the Ser226 (S226) residue (indirect modulation of Na_v_1.6), causing an increase in the Na_v_1.6-mediated Na^+^ current and in neuronal excitability. This combined direct and indirect modulation increases Na^+^ influx and increases overall neuronal excitability. (**C**) Modulatory effect of GSKβ-mediated phosphorylation on the K_v_7.2 channel. The active form of GSK3β directly phosphorylates the K_v_7.2 channel, thereby inactivating it. This inactivation of K_v_7.2 decreases the channel-mediated M-current, resulting in increased neuronal excitability. Note that PP2A-Bγ can counteract the inhibitory effects of GSK3β-mediated phosphorylation. GSK3β could phosphorylate FGF14 (dotted line), influencing the putative binding to K_v_7.2. (**D**) Modulatory effect of GSK3β on the K_v_4.2 channel. The active form of GSK3β directly phosphorylates the K_v_4.2 channel, thereby inactivating it. This modulatory effect of GSK3β decreases the K_v_4.2 channel-mediated A-type potassium current and results in increased neuronal excitability similar to GSK3β phosphorylation of Na_v_1.6. Created with http://BioRender.com (accessed on 12 April 2022).

**Figure 2 ijms-23-04413-f002:**
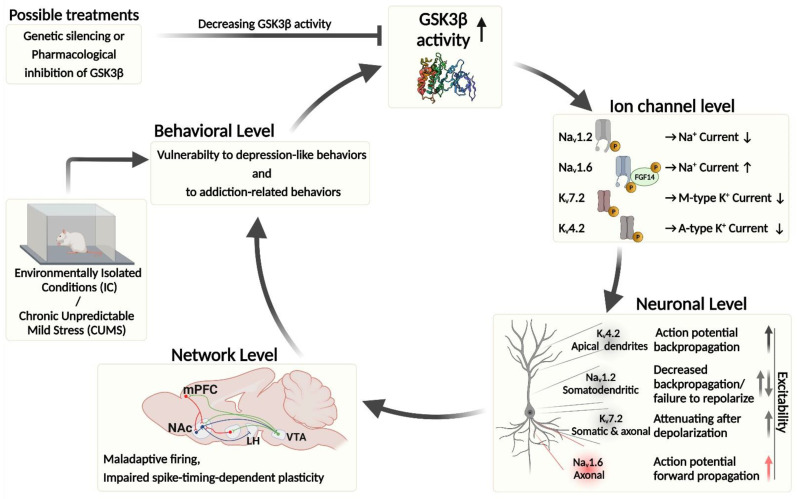
Summary of GSK3β activity on Nav channels/complexes (Na_v_1.2 and Na_v_1.6/FGF14) and K_v_ channels (K_v_7.2 and K_v_4.2) and possible outcomes at the cellular, network, and behavioral levels. Increased activity of GSK3β causes decreased Na^+^ and K^+^ currents via Na_v_1.2 and K_v_7.2/K_v_4.2 channels, respectively, while increasing Na^+^ influx via the Na_v_1.6 complex. At the neuronal level, increased activity of GSK3β leads to increased action potential forward and backward propagation via modulation of Na_v_1.6 and K_v_4.2 activity, respectively. It can also attenuate action potential repolarization via K_v_7.2 and lead to paradoxical increase in excitability via suppression of Na_v_1.2. Additionally, suppression of Na_v_1.2 induced by increased activity of GSK3β can also diminish action potential backward propagation. Schematic illustration of crucial neuronal networks (mPFC-NAc-VTA) involved in GSK3β-mediated behavioral outcomes (e.g., vulnerability to depression-like and addiction-related behavior). Environmentally isolated conditions and the chronic unpredictable mild stress model increase the activity of GSK3β and modulate the molecular-, cellular-, and network-levels of the CNS, causing maladaptive neuronal functions and serving as experimental models for testing possible treatments against the above-mentioned neuronal dysfunctions, maladaptive neuronal plasticity, and psychiatric disorders. Fibroblast growth factor—FGF; lateral hypothalamus—LH; medium spiny neuron—MSN; nucleus accumbens—NAc; medial prefrontal cortex—mPFC; ventral tegmental area—VTA; chronic unpredictable mild stress—CUMS. Created with http://BioRender.com (accessed on 12 April 2022).

## Data Availability

Data sharing not applicable.

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
