# Peer review of "Glycogen Synthase Kinase 3: Ion Channels, Plasticity, and Diseases"

_ijms, 2022, doi:10.3390/ijms23084413_

Round 1

Reviewer 1 Report

Could the authors comment on the following: given a putative action of FGF14 on Nav1.2 and its known action on Nav1.6, why do the authors rule out or do not include a similar FGF14-mediated action of GSK3b on Nav1.2 in Figure 1 and in the Nav1.2 section of the text?  

The sentence beginning on line 281 that reads "...or β-catenin knockdown decreased the levels of phosphorylated-β-catenin...."  This might be reworded to sound less redundant. 

Author Response

Responses to the comments and suggestions from Reviewer 1:

Response: We thank this reviewer for the appreciation of our work.

Could the authors comment on the following: given a putative action of FGF14 on Nav1.2 and its known action on Nav1.6, why do the authors rule out or do not include a similar FGF14-mediated action of GSK3b on Nav1.2 in Figure 1 and in the Nav1.2 section of the text?  

We have modified the figure accordingly and added corresponding text (line 343-345).

The sentence beginning on line 281 that reads "...or β-catenin knockdown decreased the levels of phosphorylated-β-catenin...."  This might be reworded to sound less redundant. 

We have reorganized the sentence. We hope we could reduce the redundancy.

Reviewer 2 Report

In their review paper entitled “Glycogen Synthase Kinase 3: Ion Channels, Plasticity, and Diseases” Mate Marosi and colleagues discuss the implications of GSK3β regulation on ion channels. The review is carefully curated and the sections are organized in a meaningful and reader-friendly manner. Figures summarize in an elegant and comprehensive manner the field and highlight the open questions regarding crosstalk between GKS3 and Na/K channels. The authors give a solid overview on GSK3 biochemistry and regulation and then focus on the connection of GSK3 to ion channels, specifically Na and K channels.

I have a general comment regarding the pharmacological approaches for GSK3β inhibition, mentioned throughout the text. I think It would be helpful for the specialized and non-specialized reader if the pharmacological approaches for inhibition of GSK3Β are specified in each section. This could be done also in the context of Figure 1 which gives an overview of GSK3-Ion channel crosstalk. As the authors state there are numerous and diverse and perhaps not so specific GSK3 inhibitors in the literature. In any case, I would propose to the authors to specify the inhibitors used in each section and perhaps comment on the validity/credibility of these inhibitors.

Specific Comments

  1. Line 65-67: This is a bit tricky sentence. It might not be entirely obvious to the reader what "negative Ser9 phosphorylation" means in this context. May be rephrase to "inhibition by Ser9 phosphorylation can override activation by Tyr216 phosphorylation". This sounds to me more straight and less ambiguous
  2. Line 156, Fig1 legend: In the text it seems that this is a direct phosphorylation event, so why in the legend the authors refer to “indirect phosphorylation” ? For example in line 255 there is a clear statement “Follow-up studies revealed that GSK3β phosphorylates FGF14 at Ser226 and that a silent Ala mutation at Ser226 reduces FGF14:Nav1.6 complex assembly”.
  3. Line 161, Fig 1 legend: correct PP2A-Bγ (gamma)

Also same line:  "inhibitory effects" might be better suited here compared to “inhibiting effects”

  1. Line 274: what "regulation... of neuronal polarity.... with ankyrin-G" means ? Please rephrase
  2. Line 297-299: this is an awkward sentence. Please correct.
  3. Line 361-362: what “heterogeneous subunits” means in this context ? Probably the authors refer to heteromeric Κv channels consisting of different Κv7 subunits?

Author Response

Reviewer 2:

Pharmacological approaches for GSK3β inhibition:

I have a general comment regarding the pharmacological approaches for GSK3β inhibition, mentioned throughout the text. I think It would be helpful for the specialized and non-specialized reader if the pharmacological approaches for inhibition of GSK3Β are specified in each section. This could be done also in the context of Figure 1 which gives an overview of GSK3-Ion channel crosstalk. As the authors state there are numerous and diverse and perhaps not so specific GSK3 inhibitors in the literature. In any case, I would propose to the authors to specify the inhibitors used in each section and perhaps comment on the validity/credibility of these inhibitors

We thank the reviewer for this comment. We have added the specific GSK3β inhibitors to the corresponding sections, but were not able to condense such information in Figure 1 while keeping the figure simple and readable. Specific details on each inhibitor can be found in the original scientific papers.

Specific comments:

  1. Line 65-67: This is a bit tricky sentence. It might not be entirely obvious to the reader what "negative Ser9 phosphorylation" means in this context. May be rephrase to "inhibition by Ser9 phosphorylation can override activation by Tyr216 phosphorylation". This sounds to me more straight and less ambiguous

We thank this reviewer for the feedback, we have reworded the sentence.

  1. Line 156, Fig1 legend: In the text it seems that this is a direct phosphorylation event, so why in the legend the authors refer to “indirect phosphorylation” ? For example in line 255 there is a clear statement “Follow-up studies revealed that GSK3β phosphorylates FGF14 at Ser226 and that a silent Ala mutation at Ser226 reduces FGF14:Nav1.6 complex assembly”

We thank this reviewer for the feedback. We have clarified the figure legend.

  1. Line 161, Fig 1 legend: correct PP2A-Bγ (gamma)

We have fixed this (PP2A-Bγ)

Also same line:  "inhibitory effects" might be better suited here compared to “inhibiting effects”

Word has been changed according to the suggestion.

  1. Line 274: what "regulation... of neuronal polarity.... with ankyrin-G" means ? Please rephrase

Sentence has been rephased.

  1. Line 297-299: this is an awkward sentence. Please correct.

This is the new sentence:” Nav1.6 channel complex; iii. lastly, the combination of effects induced by pharmacological inhibition or genetic silencing of GSK3β on the biophysical properties of the Nav1.6 channel (V1/2 of steady-state inactivation) indicates that the effect of GSK3β on Nav1.6 includes complex effects on the channel kinetics.”

  1. Line 361-362: what “heterogeneous subunits” means in this context ? Probably the authors refer to heteromeric Κv channels consisting of different Κv7 subunits?

Fixed.

“Within neurons, Kv7.2 and Kv7.3 channels exist as heteromeric and homomeric subunit assemblies that mediate M-currents - a hyperpolarizing current that reduces neuronal excitability”